# Static and Dynamic Hand Gestures: A Review of Techniques of Virtual Reality Manipulation

**DOI:** 10.3390/s24123760

**Published:** 2024-06-09

**Authors:** Oswaldo Mendoza Herbert, David Pérez-Granados, Mauricio Alberto Ortega Ruiz, Rodrigo Cadena Martínez, Carlos Alberto González Gutiérrez, Marco Antonio Zamora Antuñano

**Affiliations:** 1Engineering Departament, Centro de Investigación, Innovación y Desarrollo Tecnológico de UVM (CIIDETEC-Querétaro), Universidad del Valle de México, Querétaro 76230, Mexico; oswaldo.herbert@hakastudio.com; 2Engineering Departament, Centro de Investigación, Innovación y Desarrollo Tecnológico de UVM (CIIDETEC-Coyoacán), Universidad del Valle de México, Coyoacán 04910, Mexico; a340348289@my.uvm.edu.mx (D.P.-G.); mauricio.ortegaru@uvmnet.edu (M.A.O.R.); 3Postgraduate Departament, Universidad Tecnológica de México (UNITEC), México City 11320, Mexico; rocadmar@mail.unitec.mx

**Keywords:** dynamic gesture detection, static gesture detection, virtual reality haptics

## Abstract

This review explores the historical and current significance of gestures as a universal form of communication with a focus on hand gestures in virtual reality applications. It highlights the evolution of gesture detection systems from the 1990s, which used computer algorithms to find patterns in static images, to the present day where advances in sensor technology, artificial intelligence, and computing power have enabled real-time gesture recognition. The paper emphasizes the role of hand gestures in virtual reality (VR), a field that creates immersive digital experiences through the Ma blending of 3D modeling, sound effects, and sensing technology. This review presents state-of-the-art hardware and software techniques used in hand gesture detection, primarily for VR applications. It discusses the challenges in hand gesture detection, classifies gestures as static and dynamic, and grades their detection difficulty. This paper also reviews the haptic devices used in VR and their advantages and challenges. It provides an overview of the process used in hand gesture acquisition, from inputs and pre-processing to pose detection, for both static and dynamic gestures.

## 1. Introduction

Civilizations throughout history have shown that gestures are the easiest method of communication, surpassing even speech, mainly because they are easily understood and widely accepted, even in very different cultures. By definition, a gesture is a visual communication method (not including sound) that can be performed with the entire body or partially, e.g., the fingers, hands, face, etc. [1,2], with hand gestures being the most used. Static hand gestures can quickly communicate simple (such as numbers and letters) and complex ideas. Dynamic hand gestures are used to transmit more complex ideas, whereby deaf people can communicate complicated ideas extremely quickly [1,2]. 

Vision systems for hand and face gesture detection were developed at the beginning of the 1990s, in which computer algorithms were used to find gesture patterns in static images. Current advances in sensor technology and artificial intelligence, along with the exponential growth of computing power and GPUs, have led to the development of real-time gesture recognition for applications such as video games and virtual reality [1,2].

In virtual reality (VR), digital representations of imaginary places are created by deploying a complex mixture of 3D modeling, sound effects, and sensing technology to stimulate the user, making the immersion experience more credible and exciting [3,4,5]. Static and dynamic hand gestures could play an important role in the growing VR industry to increase the immersion and manipulation capabilities in digital scenarios and, in the case of haptic devices, to introduce a sense of touch [5,6,7].

In this paper, we present state-of-the-art hardware and software techniques for hand gesture detection, focusing on virtual reality applications. First, we introduce the general aspects that make hand gesture detection challenging, classify gestures as static and dynamic, and grade the detection difficulties. In Section 2, we define the working space for most virtual reality setups in visual hand detection techniques and the most commonly used hardware. In Section 4, we review the haptic devices used in VR applications as well as their advantages and disadvantages. In Section 5, we show a simplified process most systems use for static and dynamic hand gesture acquisition, from inputs, pre-processing, and algorithms to pose detection. Then, we analyze the most used and successful algorithms. Finally, we discuss the use of visual vs. haptic hand gesture detection, emphasizing future applications [8].

## 2. Characteristics of the Gestures

To start our analysis of gesture detection techniques, it is necessary to understand the hands’ morphology and how it influences their detection. In Figure 1, the name of the hands is shown as a type of view, and the detection mode is seen in a first-person view, which is the mode of sight in VR simulations. Most hand detection techniques detect the palmar area of the hands, while VR systems mostly analyze the dorsal area [9].

Figure 1 shows the areas of the human hand. Static gestures are mostly used to issue a specific message or state. Dynamic gestures are commonly used to express actions or more complex ideas. In addition, the detection of dynamic and static gestures varies with their degrees of difficulty. Figure 2 presents the more common static gestures and their detection difficulties. We assigned gestures 1, 2, 3, and 8 a low detection complexity since the techniques that detect the union of the fingers of the hand are not difficult. We assigned gestures 5, 6, and 7 a medium detection difficulty degree since the detection patterns produce false positives due to the similarity of the hand positions, e.g., gesture 7 communicates the message that everything is fine. It is difficult because the other fingers are occluded, and another finger (such as the pointer finger) could be in this position instead (gesture 8). Finally, we assigned gestures 9, 10, 11, and 12 a high detection difficulty degree because of the techniques applied to obtain optimal results, the occlusion of the hands, and the camera not detecting depth [8,9,10].

Figure 3 shows the most common dynamic gestures and their difficulty degrees. These kinds of gestures mostly present a continuous action, such as a greeting (gesture 2) or the gripping of objects (such as gestures 4, 5, and 6). Unlike static gestures, dynamic gestures require a more robust algorithm that allows the pattern to be detected in real-time, as well as filters to eliminate adjacent noise to avoid false positives or the generation of gestures not made by the user. Gestures 1, 2, and 3 have a low detection complexity since the hand can be detected entirely and has no occlusion. The complexity of gestures 4, 5, and 6 rises to medium since the hand hides the majority of the fingers. This type of gesture, defined as “grip-type grip and tubular-type grip”, is considered fundamental for humans to use tools or daily utensils [11,12,13,14,15,16,17,18]. Gesture 7 in Figure 3 presents two terminations and is one of the most complex dynamic gestures to detect since it shows occlusion of the hands, obstructing all the information from the camera.

**Figure 2 sensors-24-03760-f002:**
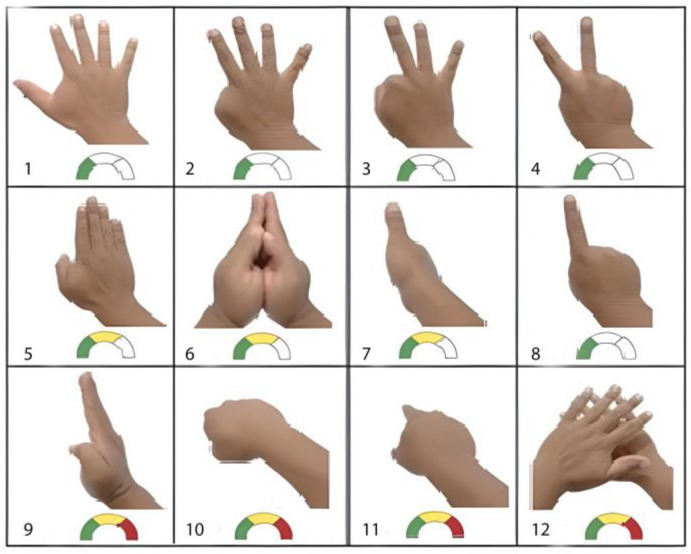
Static gestures in different forms or messages, including the detection complexity indica-tors for visual detection systems: (**1**) Dorsal front with separated fingers or number five. (**2**) Number four. (**3**) Number three. (**4**) Number two or expression of love and peace. (**5**) Dorsal front with closed fingers. (**6**) Open hands together in a vertical–transverse way or in prayer. (**7**) Closed hand with lifted thumb. (**8**) Number one. (**9**) Open hand in vertical–transverse shape. (**10**) Clenched fist. (**11**) Index finger pointing or touching object. (**12**) Dorsal front with inclination, one hiding others with separated fingers [19].

## 3. Visual System

Users have a limited area of view in different virtual reality systems according to the detection devices. Therefore, the size of the objects in the simulation and the distance between the user and these virtual objects must be correct to avoid distortions or inconsistencies in the virtual environment [20,21,22,23,24,25,26,27,28,29]. As shown in Figure 4, commercial virtual reality systems can use a screen system or mobile phone to generate a 3D-simulated virtual world.

Figure 5a shows a 3D model of a person using a virtual reality lens. It presents the volume where the system can detect the user’s gestures. Figure 5b shows the user’s hands from an aerial perspective to observe the range horizontally. The opening range from the center to the edges is approximately 62 for each eye. The field of view can be as far as the simulation allows; however, the range of interaction with objects does not exceed 1 m from the user’s perspective. Figure 5c shows a user-side view to indicate the range of vertical vision openings. In this segment, the opening range of the upper part of the horizon line is 50 and that of the lower part is 70, (b) The maximum interaction range is 1 m since the user tends to perform actions with flexed arms.

This work emphasizes these commercial systems due to their increasing use for scientific and commercial developments, continuously improving their characteristics to offer users better experiences. Although some of these systems include gestures in virtual environments, they have limitations. Examples of gesture and interaction detection systems in virtual environments are shown in Table 1. They use common optical systems, such as LeapMotion, Kinect, and mobile phone cameras, to obtain images. Their main difference is in their operation ranges, Kinect with 2.5 m and Leap Motion with only 0.40 m. However, Leap Motion offers an optimized gesture detection algorithm including finger detection. Meanwhile, hybrid systems use vision systems and controls with inertial sensors, such as Oculus Touch, HTC VIVE controllers, and Play Station Bundle. These systems have a high accuracy; however, they recognize only the hands’ position and some interactions with the buttons and fail to detect complete finger gestures [22,23,24,25,26,27,28]. 

## 4. Haptic System

Haptics refers to the ability to touch and manipulate objects based on tactile sensations, which provides awareness of stimuli on the body’s surface [30,31,32,33]. These features make haptic systems ideal for the control and manipulation of virtual reality environments. Haptics can also be classified by whether they provide force feedback, tactile feedback, or proprioceptive feedback [34]. Each type of feedback provides different information about haptic stimuli, so the function and correct choice of haptics are key to their proper application. Furthermore, the interfaces can be cataloged based on their portability or support as desktop, fixed, or portable interfaces. The latter include exoskeletons and gloves, which coat the hands to emulate their movements and are the most used and developed interfaces in the scientific and commercial fields.

Table 2 shows the most representative haptic devices according to their technique, the hardware used, or the commercial model. It describes the methods for tracking the degrees of freedom, the range of operation, and the degree of immersion a user feels while using these devices. A cyber glove comprises a system of small vibrators that generate sensations and emulate textures via different frequencies. However, it does not include force feedback and, thus, cannot identify contours. PHANTOM consists of a robotic arm that transmits force feedback via an opposing resistance to movement with DC motors, but the fingers cannot move independently [35]. Rice University’s project HANDS OMNI uses micro-air chambers to block the finger joints and, thus, movement. However, in addition to being invasive, the infrastructure needed to generate the air pressure makes the system expensive and difficult to use [36]. Dexmo was developed by Dexa Robotics and comprises a haptic glove-type exoskeleton that covers the hand and wrist, focusing on force feedback. The device provides a sense of grip that is similar to reality due to its mechanical structure; however, it is available only by pre-order as it is still under development [37].

Visual systems cannot detect some dynamic gestures due to occlusion. In these cases, haptic devices could be a good option [38]. In addition, visual systems cannot offer force feedback, which is a fast and primitive method of communication needed in the real world. The skin is the largest human organ and is full of sensors [39]. One can live without sight, but living without the sense of touch is extremely difficult, even for walking or holding objects [40,41,42]. Therefore, haptic systems for manipulating virtual environments could be essential for a realistic immersion, and more efforts should be made to develop systems that can detect more complex dynamic gestures (including position and force) [43].

## 5. Methodology

### 5.1. Gesture Detection Process for Virtual Interfaces

The process of detecting, capturing, and processing a gesture can be defined as follows. Figure 6 shows the procedure according to the method of obtaining information, using either visual (a camera, Kinect, or a smartphone) or mechanical (data gloves or Exos) systems. However, some systems use both visual and mechanical methods to increase their accuracy. Visual hardware first captures an image with a camera, which is usually infrared to obtain more accurate information [44]. At this stage, the images usually present background or ambient noise, which provides information to the images and complicates gesture detection with pre-established patterns [45,46]. Therefore, noise-eliminating filters are applied. Most algorithms are applied under controlled light conditions since this is a fundamental factor in the techniques’ accuracy. After the noise is eliminated from the images, the method or algorithm for gesture detection is applied. Once the gesture location with the coordinates or positions is obtained, a 3D model of the fingers and hand is created to generate the gesture in a virtual system [34,35].

The capture methods used by data gloves or Exos increase their accuracy by being directly mounted on the hand [47]. These methods use accelerometers, gyroscopes, and resistive sensors to detect finger movements and hand positions; however, the electronic systems generate noise during movements and require many sensors to include all the fingers [48,49]. Therefore, an electronic filtering stage that eliminates the noise generated by the hardware is required. Once the information is obtained, it is compared with expected models or databases to implement the positioning algorithm for the hand parts. This method requires a lower processing capacity since the algorithms tend to be simpler than vision algorithms [34,35,50]. 

### 5.2. Gesture Detection Techniques

The problems with the different techniques include the amount of light when conducting an experiment, which is crucial to obtain a good result, and the amount of information presented in the environment or background, which usually increases the noise during gesture detection. Several works on gesture detection were published in the 1990s. The techniques were based on several classifiers for hand gesture recognition (HGR), including the k-nearest neighbors’ algorithm, support vector machines (SVMs), neural networks (NNs), and finite-state machines (FSNs), in addition to hidden Markov models and neural networks for calibration [51,52,53,54,55,56]. Table 3 and Table 4 show the detection techniques for static and dynamic gestures, respectively. Each table shows the techniques’ accuracies, main characteristics, and possible improvements according to the literature. The techniques were selected due to being highly efficient and the most used by researchers. The techniques use similar learning algorithms, such as KNN, artificial neural networks, SVMs, and CNNs, which can adapt to the tonal and morphological variations in the hands of different users. The statistical algorithms presented, such as hidden Markov models (HMMs), multi-layer perceptron, and Euclidean distance, optimize the amount of processing by evaluating the positions of the different hand parts and eliminate the errors produced by impossible positions. Table 3 and Table 4 show the probabilistic semantic network [11,12,13,14,18], RGB filter and binary mask [13,14,15,16], and distance transform [15,16,17,19], techniques that are optimized for use in virtual reality systems and allow detection employing smartphones. Despite being static gesture detection techniques, the algorithms can be optimized for detecting dynamic gestures, allowing users to use virtual reality helmets to visualize their hands in simulations and increase the degree of immersion. Performing these techniques with smartphones allows the experiments to be easily replicated and improves them. Some researchers have used the combination of a mobile phone camera and gloves with sensors to increase accuracy, generating applications for sign language and motion limiters for job training applications or medical rehabilitation [15,16,17,18,20,21,22,23,24,25,34]. Additionally, Table 5 details the techniques specifically used for dynamic gesture detection, highlighting their accuracy, main features, and required hardware, thereby providing further insight into the advancements in this area.

Four types of Hardware that are shown in Table 6 were used.

Tests were performed on the devices in Table 6. Using Kinect (Redmond, WA, USA) was ruled out due to Microsoft’s statement about terminating its production. The use of conventional cameras presented good results but was limited by needing to be connected to a PC, making the manipulation of objects difficult due to the limited area.

Figure 7 shows two images of a hand obtained with a GoPro Hero camera (San Mateo, CA, USA) in a panoramic format using a 120 °C lens aperture. A binary filter was applied to the images to obtain the area used by the hand. This camera had a 4K-type resolution, which generated a 6 s delay in the flow of information to the PC for the real-time application of the algorithm, so it was discarded.

Finally, a mobile phone camera can also be used due to the technological advancements in these devices. We explore this in the next section. 

### 5.3. Gesture Capture Using a Smartphone

Virtual reality simulations can be implemented using cardboard virtual reality lenses, performing all the processing with a mobile phone. Figure 7 shows that the hardware used and the characteristics of the technique’s application must be established in the image capture, pre-image processing, image processing, virtual model generation, and model-checking processes. The hardware used was a Samsung S6 cell phone, and the technical specifications are shown in Table 7.

Remarkably, solutions based on machine learning and artificial intelligence have revolutionized gesture detection techniques in virtual reality (VR) environments. These technologies enable precise and rapid recognition of both static and dynamic gestures, significantly enhancing human–computer interaction [59,67]. For instance, Shantakumar proposed a method based on angular velocity that achieves efficient real-time gesture recognition without the need for extensive data preprocessing [68]. This approach, supported by machine learning algorithms, has demonstrated high accuracy in gesture detection, making it ideal for high-frequency interactive applications such as video games and interactive tools [69].

Moreover, the implementation of machine learning in gesture detection in VR environments has allowed for overcoming previous limitations related to self-occlusion and complex finger movements [70,71]. By utilizing motion tracking sensor-based systems, precise capturing of three-dimensional hand and finger movements is achieved, thus avoiding common issues in vision-based systems [68]. This enhanced recognition capability, supported by artificial intelligence algorithms, has paved the way for more natural and fluid interaction in VR environments, providing users with an immersive and engaging experience [72].

### 5.4. Image Capture

An open-source system that allowed access to the cell phone camera to capture and process a video was used. In the first test, a hand-shaped pattern with a black glove was taken as a reference to make the negative. The code was based on shape detection and programmed for Android in C. However, it generated considerable errors when detecting other black areas, so the use of a glove was ruled out. It was established that a start calibration pattern was needed to detect the contrasts in the user’s hand using the detection system. Figure 8 shows the figures obtained with a GoPro Hero3 camera, with a binary filter applied.

Figure 9 shows the image capture process, wherein a few hands were taken as examples to determine the hand’s total area without defining the fingers.

Figure 10 shows the input images using the smartphone’s rear camera.

The relevant techniques and filters were applied via this pre-processing. The location of the hands could be detected in the total capture space of the smartphone camera used, so the first algorithm optimization was performed by eliminating the areas that did not require analysis.

### 5.5. Application of Filters

Once the area to work was obtained, RGB or color filters were applied, as they only needed to be applied to the area where the hand was located. The RGB characteristics and value extraction process are shown in Figure 11 and Figure 12.

The descriptors extracted while capturing the RGB values of the capture area where the hand was located allowed us to obtain the nuances of the user’s hand and the RGB values of the background and other elements, such as clothing. Once the skin’s RGB value was captured, the parts of the image in which the skin color was found could be detected to delimit it from other values. Figure 12 shows the application of the binary filter, in which only the skin’s RGB value is assigned a positive value or a value of one, while everything else is assigned a negative value or a value of zero to obtain a negative contour.

This process resulted in a high gesture detection accuracy under optimal conditions with a regular light level, neither saturated nor absent. Figure 13 shows the results of the filter’s application under different light levels, whereby a problem could be detected without adequate light.

The solution was to generate a new descriptor to perform a calibration based on the hand’s skin tone.

New descriptor: The step to obtain the chromatic coordinates (intensity division) was
$$\left(\mathrm{RGB}\right)\to \left(\frac{R}{R+G+B}+\frac{G}{R+G+B}+\frac{B}{R+G+B}\right)$$

The change in the image intensity, which is a scalar product, was defined as follows:RGB→s·RGB→sR·sG·sBs∈ ℝ

The intensity was canceled, and the new descriptor was invariant to the intensity:$$\left(\mathrm{sRsGsB}\right)\to \left(\frac{\mathrm{sR}}{\mathrm{sR}+\mathrm{sG}+\mathrm{sB}}+\frac{\mathrm{sG}}{\mathrm{sR}+\mathrm{sG}+\mathrm{sB}}+\frac{\mathrm{sB}}{\mathrm{sR}+\mathrm{sG}+\mathrm{sB}}\right)=\left(\frac{\mathrm{sR}}{s\left(R+G+B\right)}+\frac{\mathrm{sG}}{s\left(R+G+B\right)}+\frac{\mathrm{sB}}{s\left(R+G+B\right)}\right)$$

The new descriptor was applied to obtain the chromatic coordinates, which were independent of the amount of light in the experiment. As shown in Figure 14, the new descriptor was applied to the images with light variation, which obtained images with chromatic coordinates and eliminated the problem of light incidence.

## 6. Results

The techniques in this paper were studied according to the detection of static or dynamic gestures, the types of applications, and the mechanical (gloves or armbands) or visual (cameras or infrared sensors) systems used. Mechanical systems usually have greater precision but require being in contact with the user, reducing their comfort and portability. Some systems have become commercial, for example, Dexmo and ManusVR can be purchased online since the other methods are under development in laboratories, making it difficult to replicate the reported results [1,2,5,7,9].

Visual techniques use a camera system, such as Kinect [10,11,12,13,14,15,16,17,19,20,21,23,34]. Although Kinect has been used thanks to the publication of its codes and functionalities, Microsoft replaced this device in 2017, with Azure Kinect, which was discontinued in 2023. Nevertheless, the technique can be applied to other types of cameras. Kinect was developed for use in infrared sensors. Therefore, several researchers have used a similar system called LeapMotion, although its range of coverage is lower [73].

Table 3 shows the RGB color filter and the binary filter [22,23,24,25,26,30,31,74], techniques (although they could also be included in Table 4 since they require little processing capacity). These easily applied techniques can detect dynamic gestures via mobile devices and are used in most of the techniques shown in Table 3 and Table 4 to reduce environmental noise. The limitation of the RGB color filter and the binary filter is in the hand’s morphology. If the camera does not detect the correct form, the filters will take the hand’s shape as noise, thus not presenting information for image processing. Therefore, the RGB color filter and the binary filter are added to other techniques that allow the hand to be detected when blocked by objects or superimposed hands. The techniques shown in this paper are divided according to the detection of static and dynamic gestures and the type of application to which they are applied. However, they can also be divided into mechanical (use of gloves or bands on the arm) and visual (use of cameras and infrared sensors) systems. The mechanical ones usually have greater precision, but they need to be in contact with the user, reducing the comfort and in many cases the portability of the systems. Few systems have become commercial, e.g., GloveOne, Dexmo, and ManusVR are the only ones that can be purchased online since other methods are laboratory developments of which the reported results make them very difficult to replicate [1,2,5,7,9,26,34,35].

The challenge of detection lies in the color detection calibration at the beginning of the image capture and filter’s application. Figure 15 shows the results of a mobile application where image capture, chromatic descriptions, and the binary filter were implemented. First, the user positioned his/her hand on the device, allowing the RGB value to be captured. (The application of a neural network allowing the self-calibration of the hand’s RGB values is proposed for this method in future works). The chromatic detection model was then applied, whereby the light values were eliminated to obtain only the values of the hand with the chromatic descriptions. The binary filter was applied with the obtained values, and the saturation values were added to simplify detection. However, increasing this range required a greater processing capacity.

Figure 16 shows the mobile application that allows the hand position to be detected in motion after applying all the necessary processes.

### Virtual Interface Results

The virtual interface was developed using the Unity 3D program as it supports most 360 virtual vision lenses. This project used the Oculus Rift V3 system. Figure 17 shows the graphical interface of the software.

As the image shows, the software allows the programmer to have a development view and a view window with dual focuses, which is the view in each lens of the Oculus Rift viewfinder (Irvine, CA, USA). The image undergoes some distortion due to the lenses of the virtual reality viewer. As shown in Figure 18, the user can visualize their hands via the smartphone camera. In the environment selected for the application, the user needed to touch buttons and doors to advance through a corridor where he/she interacted with 3D devices that enabled him to identify that his fingers were interacting with a 3D object.

The physical characteristics of objects can be replicated in the virtual environment of Unity software. The virtual hand was designed to represent gestures, based on the morphology of a real hand. The 3D model of the hand presents divisions with phalanges to show the movement of each finger of the two hands. In addition, a virtual space can be generated in which the user interacts with objects in the capture range of the mobile phone’s camera.

## 7. Conclusions and Discussion

### 7.1. Discussion

Gesture recognition can be seen as the way in which computers interpret human body language, being considered a natural human–computer interface. In a gesture recognition system, a gesture model appropriate to the context and application is initially defined, which in turn allows defining the interactions between specific applications and the proposed architecture [61]. However, due to the number of possible movements that can be executed by the human body, it is important to determine the types of gestures that are analyzed and recognized. Taxonomies that organize and delimit the types of gestures to be analyzed, executed, and employed have already been proposed [62,63].

Human–computer interaction seeks to make the use of computers easier, more intuitive, and more comfortable. In general, it studies the design and development of new hardware and software interfaces that enable and improve the user’s interaction with the computer [61,62]. In this context, Microsoft’s Kinect has become a widely used device in this area, as it provides researchers and developers with a large amount of spatial information of objects in a real scene. This device has enabled the development of multiple systems for educational and entertainment video games, physical rehabilitation, robotic and computational control, and augmented reality [59,62,63,64,65,66,67]. Performing interactions through a device of this type may require various tasks such as finding and identifying a user, recognizing the parts of his/her body, and the recognition of gestures (the indications) that he/she makes, among others. Of particular interest for this research is the task of detecting the hand and recognizing gestures made with the fingers, in particular the gesture of touching [59,65,66,67,68].

In the state-of-the-art we can find several works focused on both hand detection and gesture recognition. This is due to the naturalness and intuitiveness it offers, besides being a striking mechanism, which motivates people to use it. However, to perform a proper detection of gestures, specialized hardware is being used, which can be difficult to access due to costs and infrastructure. In this work, we propose a gesture detection strategy using webcams (which are non-specialized hardware easily accessible in a standard computer), where through image preprocessing the noise is reduced, and using classifiers such as support vector machines, the detection of the gesture made by the user is performed [59,64,65,66,67].

In recent years, hand gesture recognition has become a useful tool in the process of human-–computer interaction (HCI). HCI seeks to transfer the naturalness of the interaction that occurs between people to the interaction between humans and computers, interaction that has been conditioned and limited by the use of mechanical interfaces and devices such as the mouse and keyboard [46,47]. Several studies, showing different architectures based on gesture recognition, present different ways of developing systems for recognition and their integration with specialized and non-specialized hardware [50,51,52,53,54]. Gestural interaction applied to technology is increasingly part of our daily life through mobile devices [58,73,74].

In this area, the best known interaction is currently performed with gestures on touch surfaces. The evolution of touch technology has meant that we now have multi-touch screens, which allow the recognition of different points of contact through pressure and gestural interaction. There are a multitude of touch interaction gestures; the best known are those based on one touch, double touch, and continuous touch to make a selection, join two fingers to scroll, or join two fingers and extend them to perform a zoom effect [68,69,70,71].

During the development of this work, it was established that an adequate localization of the region of interest is fundamental in a gesture recognition system, as the accuracy in the detection of the area of interest directly influences the characteristics obtained by the extraction process, and consequently, the results provided by the classification method.

### 7.2. Conclusions

The analyzed research presents the results for specific hand positions, but virtual reality users visualize their hands conversely. Therefore, the results should be presented based on the gestures generated by virtual reality users and show the degree of efficiency in achieving both static and dynamic gestures. The described techniques present a degree of efficiency for specific light and background conditions. However, algorithms must be designed to withstand general conditions to be implemented in applications with varying conditions and be highly efficient to achieve an acceptable level of user immersion. According to the techniques’ complexities, most were shown to use color and binary filters to eliminate noise. The most efficient techniques are based on statistical models, using intelligence computing algorithms to calibrate the capture ranges due to the variations in hand morphologies and colors. The techniques must be optimized to detect gestures dynamically with higher efficiency using mobile devices due to the demands of virtual reality applications, where greater accuracy in locating the hands and fingers is required.

Additionally, the combination of hand gestures with haptic devices can significantly enhance the virtual reality experience. Haptic devices provide tactile feedback, allowing users to feel virtual interactions more realistically. This improves the accuracy of gesture detection and increases user immersion by providing tangible responses to virtual actions.

For example, using haptic gloves such as the Dexmo or ManusVR in conjunction with gesture recognition techniques can achieve more precise finger movement detection and better interaction with virtual objects. These devices can be calibrated to work in tandem with RGB and binary filters, adjusting detection parameters in real time based on the haptic feedback received.

The integration of these systems addresses variations in hand morphologies and lighting conditions and provides a robust platform for complex VR applications, such as precise object manipulation and the simulation of physical interactions in a virtual environment. This combination of technologies is crucial for the development of advanced virtual reality applications that require high precision and an immersive, realistic user experience.

## Figures and Tables

**Figure 1 sensors-24-03760-f001:**
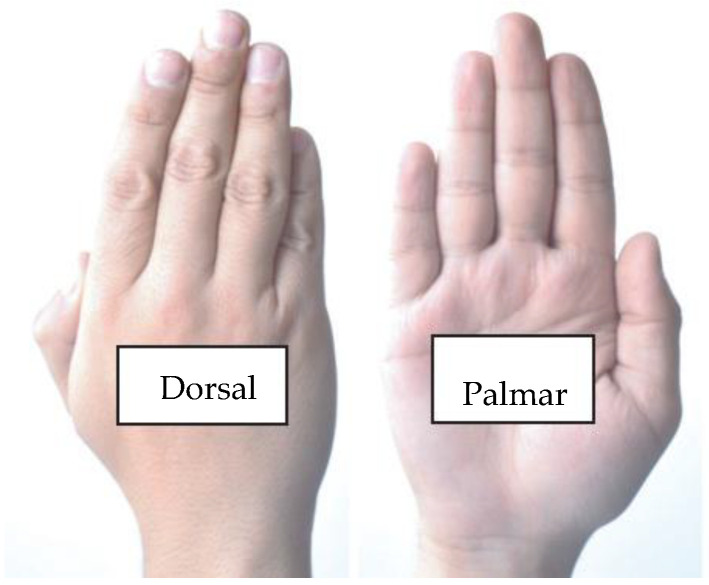
The dorsal and palmar hand areas.

**Figure 3 sensors-24-03760-f003:**
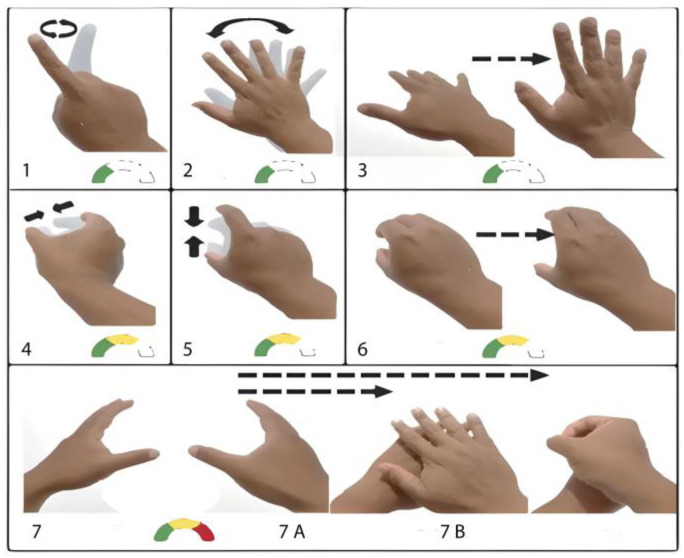
Dynamic gestures in different forms or messages, including the detection complexity indicator for visual detection systems: (**1**) Closed hand with index finger indicating a circular shape. (**2**) Dorsal face with separated fingers for greeting. (**3**) Dorsal front with separated fingers with forward movement to touch an object. (**4**) Thumb and index finger separated for vertical grip. (**5**) Thumb and index finger separated for horizontal grip. (**6**) Closed hand to hold a horizontal tubular shape. (**7**) A: Hands separated to envelope one another in an open face with separated fingers. (**7**) B: Hands separated to grip with both.

**Figure 4 sensors-24-03760-f004:**
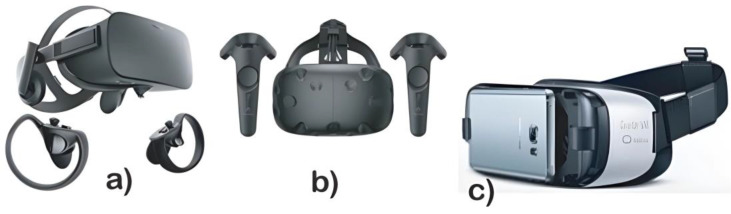
Commercial head-mounted displays for virtual reality. (**a**) Oculus Rift, (**b**) HTC VIVE, and (**c**) Samsung Gear; this headset uses a mobile phone to reproduce the virtual reality, and it is similar to Google Cardboard [29].

**Figure 5 sensors-24-03760-f005:**
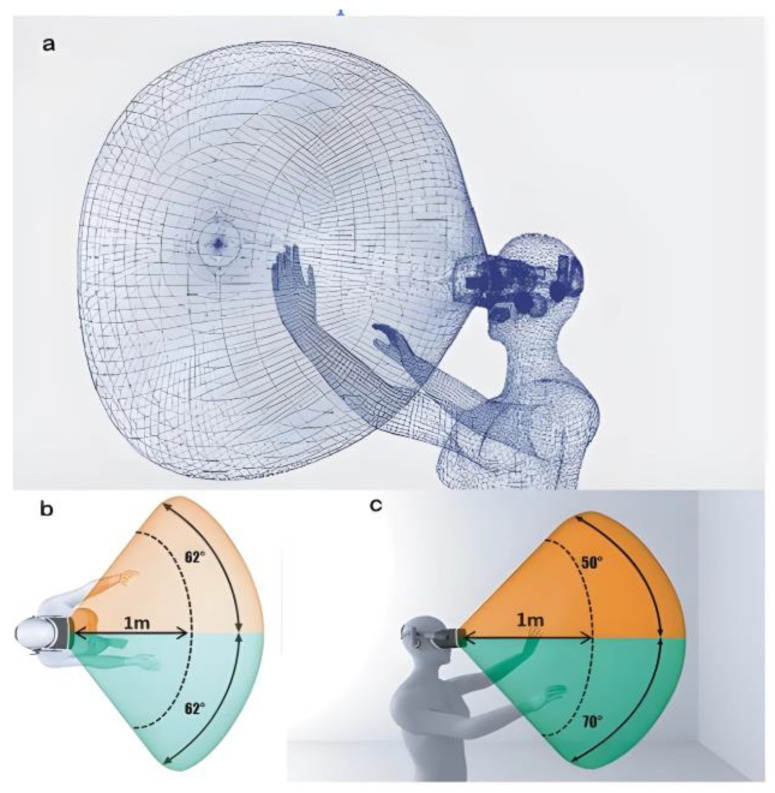
(**a**) Vision volume in virtual reality systems with hand gestures. (**b**) The range of horizontal vision. The optimal range of work is 1 m, in commercial virtual reality systems. (**c**) The range of vertical vision.

**Figure 6 sensors-24-03760-f006:**
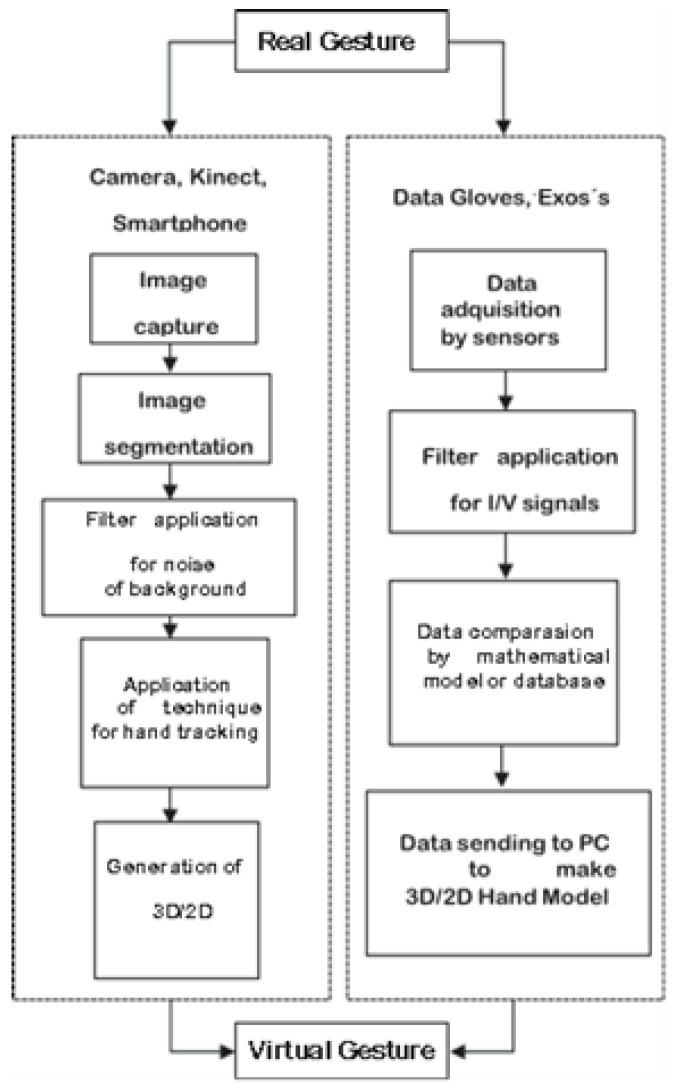
Diagram of the gesture detection process using different types of hardware (visual or nonvisual).

**Figure 7 sensors-24-03760-f007:**
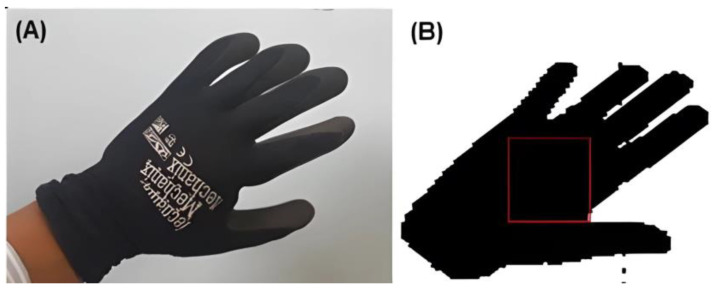
Glove pattern detection using a cell phone camera. The use of this specific glove is required, reducing the possibility of using other devices. (**A**) is the original photograph taken by a mobile camera, it is shown the glove used to generate a difference in the color pattern. (**B**) is a binary B&W image used as a monochromatic change detection, it only detected a rough approximation. (**A**) is the original image taken by the cell phone camera, showing the glove used to create a difference in the color pattern. (**B**) is the image after applying a binary filter for monochromatic color change detection, where only the color difference is detected but the shape is not correctly identified.

**Figure 8 sensors-24-03760-f008:**
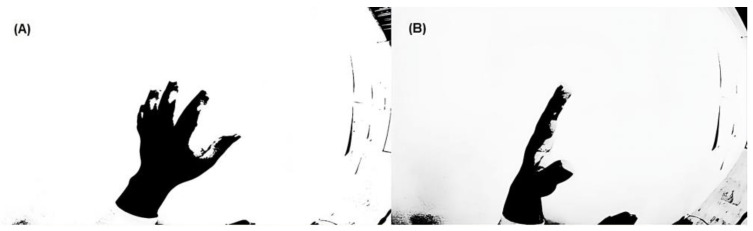
Images were obtained from a GoPro Hero3 camera, with a binary filter applied. (**A**) was obtained after a binary filter using a black glove over a white background, for gesture detection. (**B**) was captured in same way after a rotation. (**A**) shows an image obtained by applying a binary filter to a hand wearing a black glove on a white background for gesture detection. (**B**) shows the same image after a rotation, neutral grip at 90° to the horizon.

**Figure 9 sensors-24-03760-f009:**
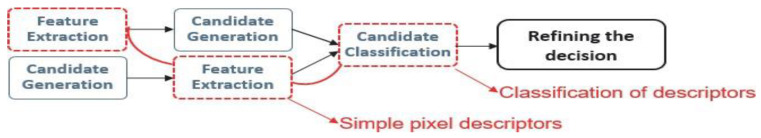
Image capture process.

**Figure 10 sensors-24-03760-f010:**
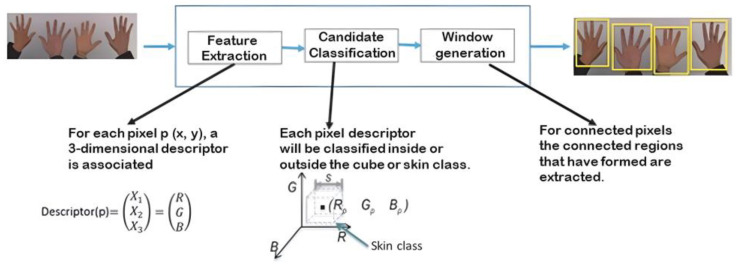
Object detection diagram of the total occupied area without finger definition.

**Figure 11 sensors-24-03760-f011:**
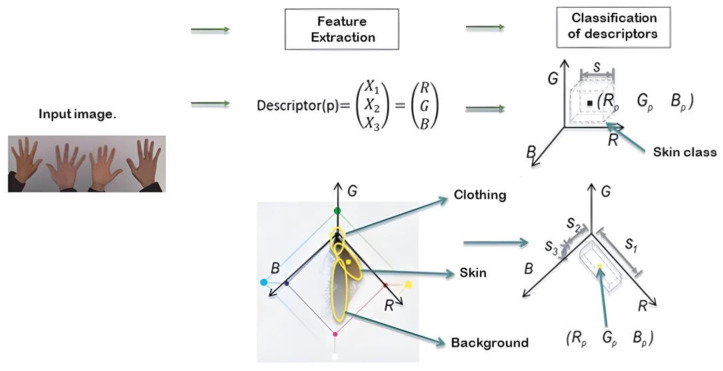
RGB descriptor extraction process for the classification of areas such as clothing, skin, and the background.

**Figure 12 sensors-24-03760-f012:**
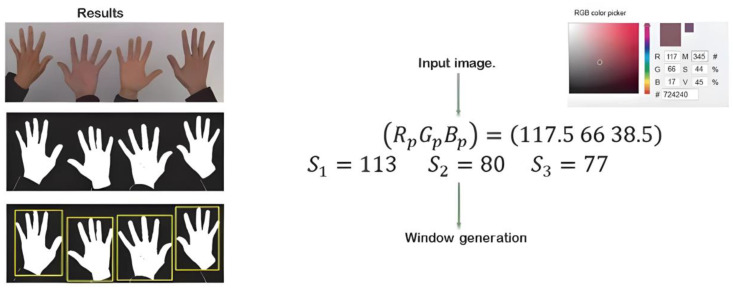
Binary filter application process.

**Figure 13 sensors-24-03760-f013:**
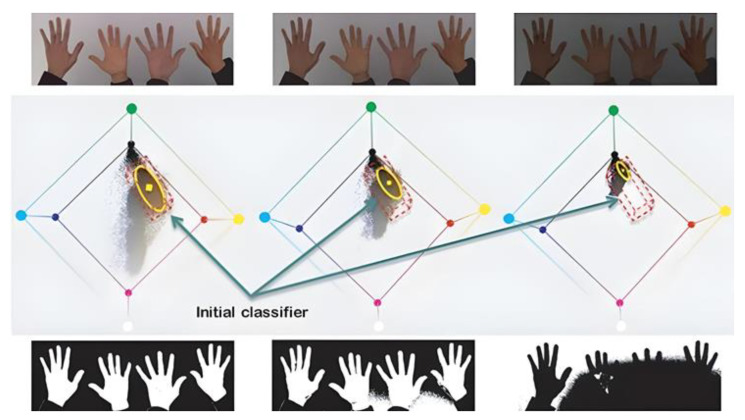
Binary filter’s application under three light levels.

**Figure 14 sensors-24-03760-f014:**
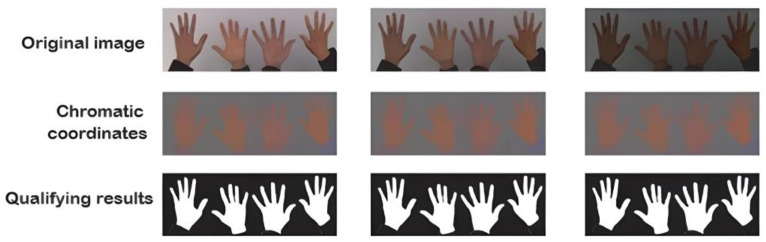
Results obtained by applying a new descriptor before applying the binary filter.

**Figure 15 sensors-24-03760-f015:**
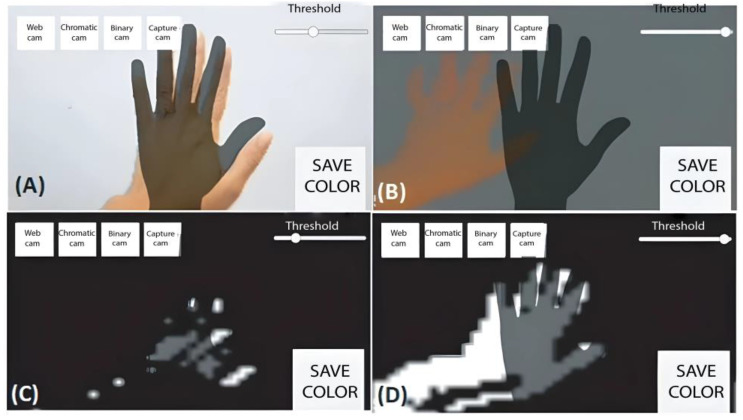
Mobile application for detecting the hand’s RGB value and the application of chromatic qualifiers and a binary filter. (**A**) depicts the color calibration and amount of information, (**B**) presents the color camera detection, (**C**) shows the detection range setting at 30%, and (**D**) illustrates the detection range setting at 85%.

**Figure 16 sensors-24-03760-f016:**
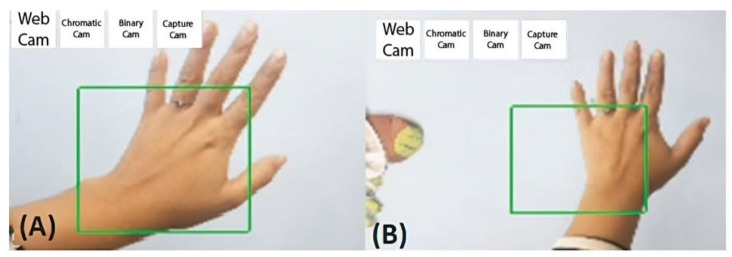
Mobile application to detect hand position. (**A**) presents the detection of the hand position without a filter. (**B**) illustrates the detection of the hand position with noise.

**Figure 17 sensors-24-03760-f017:**
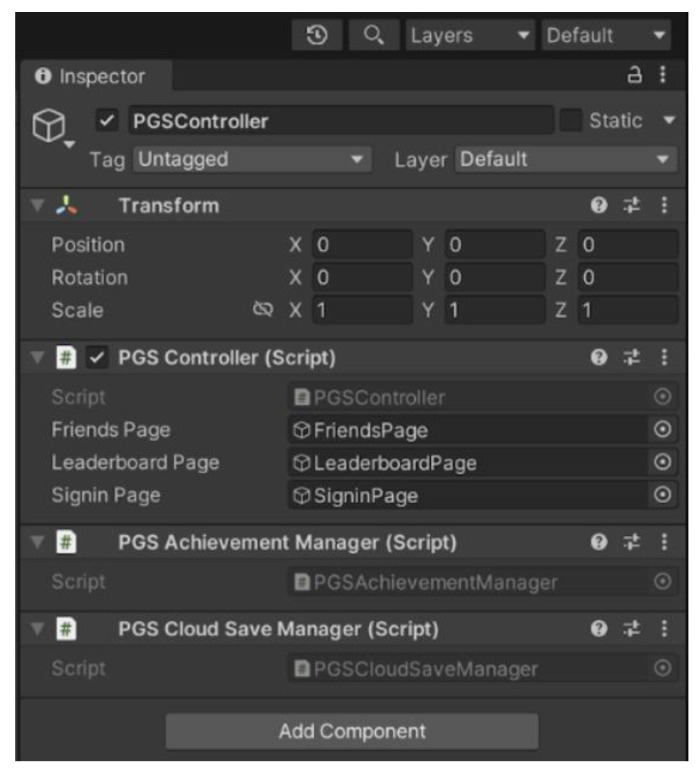
Unity 2021.3 LTS. 3D software interface.

**Figure 18 sensors-24-03760-f018:**
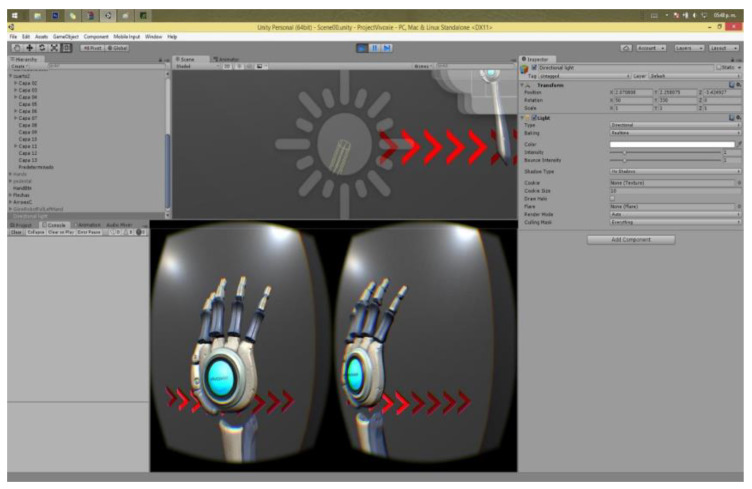
Hand modeling for interaction with the virtual application.

**Table 1 sensors-24-03760-t001:** Virtual reality systems and hand-tracking devices.

Company	Device Name	Tracking	SDK	Range (m)	Weight (g)	Grade of Immersive	Status
Microsoft	Kinect	Infrared projector and RGB camera	Tes	2.5	430	++	On Sale
-	Camera WEB	Camera	Depends on the algorithm	1	100–400	+	On Sale
LeapMotion (San Francisco, CA, USA)	LeapMotion	Two monochromatic IR cameras and three infrared LEDs	Yes	0.4	4+	+	On sale
Mobile phones	Mobile camera	Camera	Depends on the algorithm	0.6	Depends on the mobile phone	+	On sale
Facebook (Cambridge, MA, USA)	Oculus Touch	Infrared sensor, gyroscope, and accelerometer	Yes	2.5	156	++	On sale
HTC (Taoyuan City, Taiwan)	HTC Vive touch	Lighthouse (2 base stations emitting pulsed IR lasers), gyroscope	Yes	3	160	++	On sale
Sony (Tokyo, Japan)	PlayStation VR bundle	Two-pixel cameras with lenses	No	2.1	120	++	On sale

**Note:** An important feature is immersiveness, it seeks to replicate the real, physical world through a digitized experience. It has no defined range, “+” indicates less immersive and “++” more immersive.

**Table 2 sensors-24-03760-t002:** Comparison of commercial and non-commercial haptic interfaces for virtual reality.

Autor/Company	Device Name	Type	Tracking	DOF (Degrees of Freedom)	SDK	Range (m)	Weight (g)
Cyber Glove System (San Jose, CA, USA)	Cybergrasp	Exo	Accelerometers	11	No	1	450
Neurodigital technologies (Almería, Spain)	Glove one	Glove	IMU’S (accelerometers and gyroscope)	9	Yes	1	100
Manus VR (Geldrop, The Netherlands)	MANUS VR	Glove	IMU’S (accelerometers and gyroscope)	11	Yes	2	100
Dexta Robotics (Shenzhen, China)	DEXMO	EXO	IMU’S (accelerometers and gyroscope)	11	Yes	2	190
Sensable Robotics (Berkeley, CA, USA)	PHANToM	Robotic arm	Encoders	2–6	Yes	-	1786
Mounrad Boutzit, George Popescu, Grigore Burdea and Rares Boian, 2002	Rutgers Master II ND	Glove	Flex and hall effect sensors	5	No	2	80
Robot Hand Unit for Research	HIRO III	Robotic arm and glove	Encoder	15 for the hand, 6 for the arm	No	0.09 m^3^	780
Vivoxie (Ciudad de México, Mexico)	Power Claw	Glove	Monochromatic IR cameras and infrared LEDs (Leapmotion)	12	Yes	650 mm^3^	120

**Table 3 sensors-24-03760-t003:** Static gesture detection techniques.

Technique	Accuracy (%)	Researchers	Advantage	Hardware
Bhattacharyya divergence into Bayesian sensing hidden Markov models	82.08–96.69	Sih-Huei Chen et al., 2017. Kumar, 2017 [56]	Ability to deal with probabilistic features	Kinect
SHREC 2021	95–98	Caputo et al., 2021 [57]	Recognition of 18 gesture classes	Camera
Artificial Neural Network with Data fusion	98	Ali et al., 2023 [58]	The ANN achieved increased recognition accuracy	Kinect
Manus Prime X data gloves, data acquisition, data preprocessing, and data classification to enable non-verbal communication within VR.	95	Achenbach et al., 2023 [59]	Data acquisition, data preprocessing, and data classification to enable nonverbal communication within VR	Meta-classifier (VL2)
The multi-scale CNN employs LSTM to extract and optimize temporal and spatial features. A computationally efficient SBI-DHGR approach	90–97	Narayan et al., 2023 [60]	Does not need the power of processing data	The algorithm can be extended to recognize a broader set of gestures

**Table 4 sensors-24-03760-t004:** Dynamic gesture detection techniques.

Technique	Accuracy (%)	Researchers	Advantage	Hardware
Deep learning and CNN	90	Jing Qi et al. (2023) [37]	Using deep learning framework to obtain initial parameters for the optimization algorithm is effective in the sense that it allows us to correctly track complex gestures and very fast movements in the finger	Web Camera
LDA, SVM, ANN	98.33	Qi, J et al., 2018 [38]	The ANN achieved the lowest error rate and outperformed LDA and SVM in all the test	IMUd’s in Data Glove
Use of a well-designed end-to-end architecture based on 3D DenseNet and LSTM variants	92–99	Lu et al. (2023) [61]	Use of a well-designed end-to-end architecture based on 3D DenseNet and LSTM variants	Web Camera
Use of Deep Neural Models to achieve better results for both static and dynamic gestures.	95	Sarma et al. (2023) [62]	Improves segmentation accuracy.	Video
Deep learning	92	Mahmud et al. (2023) [63]	CRNN (convolutional recurrent neural networks)	Web Camera
Deep learning	99	Karsh et al. (2023) [64]	A two-phase deep learning-based HGR system reduces computational resource requirements	Web Camera

**Table 5 sensors-24-03760-t005:** Techniques used for Dynamic gesture detection.

Technique	Accuracy (%)	Researchers	Advantage/Main Feature	Hardware
Filters: Gaussian Blur, CLAHE, HSV, Binary Mask, Dilated and Eroded	87.89	Maisto, M. et al., 2017 [34]	It can be used in VR and requires little information of the captured image.	Camera and Leap Motion
Euclidean distance	84	Culbertson, H et al., 2018 [30]	No training algorithm is needed	Leap Motion and Data Glove
Deep learning	99	Miah et al. (2023) [65]	Used two graph-based neural network channels in the multi-branch architectures and one general neural network channel; high-performance accuracy and low computational cost	Web Camera
Deep learning	97	Rastgoo et al. (2023) [66]	Dynamic Hand Gesture Recognition (DHGR) Zero-Shot from multimodal data and hybrid features.	Web Camera

**Table 6 sensors-24-03760-t006:** Hardware used in research development.

Technique	Description	Advantage	Disadvantage
Kinect	Infrared projector and RGB camera	Ease of implementation algorithms, Open Source	It stopped being manufactured in 2017; it must be connected to a PC.
Conventional camera	60 fps camera	Low-cost and universal PC driver	It must be connected to a PC
GoPro	4K resolution up to 256 fps	High image quality, greater image capture range	High cost, slow process of sending information to a PC
Mobile phone camera	Device embedded in all Smartphones	Easy access to algorithm implementation, low cost, and high transfer speed.	Limited to the mobile phone processor.

**Table 7 sensors-24-03760-t007:** Samsung S6 mobile phone specifications.

Item	Specification
Screen	5.1-inch QHD SMOLED
Resolution	2.560 × 1.440 pixels
Pixel density	577 ppp
Operating system	Android
Processor	2.1 GHz Samsung Exynos (eight cores)
Camera	16 Mega-pixels
RAM	3 GB

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
