# Peer review of "Static and Dynamic Hand Gestures: A Review of Techniques of Virtual Reality Manipulation"

_sensors, 2024, doi:10.3390/s24123760_

Round 1

Reviewer 1 Report

Comments and Suggestions for Authors

The research article presents an analysis of gesture recognition techniques in the context of Virtual Reality (VR) applications, focusing on the use of smartphones and mobile devices for capturing and processing hand movements. The research provides an overview of various methods and technologies, including mechanical and visual systems, and evaluates their effectiveness in achieving efficient and accurate gesture recognition in VR environments.

Shortcomings:

1. The article's use of English is somewhat inconsistent, making it difficult at times to follow the flow of the text.

2. The quality of the figures and illustrations in the article is poor, with some images appearing unclear or lacking in detail.

3. While the research does touch on potential improvements and future research, a more in-depth discussion of possible advances in gesture recognition and VR would add value to the article and provide clearer guidance for future work.

4. The article mentions the use of specific models and techniques, such as neural networks, but does not always provide sufficient detail on the implementation and evaluation of these methods.

5. References to previous work from the global scientific community (2022-24), consistently presented at video or multimodality oriented conferences (CVPR, ICCV, ECCV, ICASSP, INTERSPEECH, and others) or in first quartile journals (Sensors, ESWA, Neurocomputing, and others) need to be expanded. 

Modern methods can be found on paperwithcode, on the SOTA page of a particular case. For example, if we consider the modern, well-known and publicly available corpus (https://paperswithcode.com/sota/sign-language-recognition-on-autsl), the best methods at the moment are STF+LSTM (for mobile devices - overlapping with the review), SAM-SLR, Ensemble - NTIS, MViT-SLR, FE+LSTM (for mobile devices - overlapping with the review). Therefore, it is appropriate for the authors to analyze the best and known methods. All this will show that the authors are aware of the best results achieved on large corpora. Otherwise, it turns out that the authors only described neural network models for gesture recognition and did not mention the best results of 2022-24. The methods build on MediaPipe and other approaches and describe their application on mobile devices (STF+LSTM and others), so these methods are popular in the research community and can perfectly complement the tables in the paper. As for other corpora, for example, the HaGRID case, recently presented at the WACV 24 conference, is new and there are no SOTA results for it yet. As for the review of Kinect-related work (not only the first version, but also the second version - which is already considered obsolete), a certain number of corpora have also been collected for this purpose in the field of gesture recognition, and studies have been carried out based on them (see data from TheRusLan corpus - highly rated LREC conference) and methods.

6. In this article, the authors should clearly categorize existing approaches and methods for automatic hand gesture recognition and sign language elements into four distinct categories: (1) gesture recognition using sensor-equipped gloves, (2) marker-based motion capture systems, (3) video-based motion capture hardware, and (4) computer vision and machine learning methods. This categorization is essential to provide a comprehensive overview of the field.

Comments on the Quality of English Language

Extensive editing of English language required.

Author Response

Dear Reviewer
Thank you very much for your comments. We have tried to comply with each and every one of them.

  1. The article's use of English is somewhat inconsistent, making it difficult at times to follow the flow of the text.

R= Thank you very much for your comments, the English language work was reviewed by MDPI Editorial services.

  1. The quality of the figures and illustrations in the article is poor, with some images appearing unclear or lacking in detail.

R= Thank you very much, all the figures were improved

  1. While the research does touch on potential improvements and future research, a more in-depth discussion of possible advances in gesture recognition and VR would add value to the article and provide clearer guidance for future work.

R= Thank you very much, the suggestions were attended to. See lines 380-432

  1. The article mentions the use of specific models and techniques, such as neural networks, but does not always provide sufficient detail on the implementation and evaluation of these methods.

R= Thank you very much, the suggestions were attended to. See lines 241-257

  1. References to previous work from the global scientific community (2022-24), consistently presented at video or multimodality oriented conferences (CVPR, ICCV, ECCV, ICASSP, INTERSPEECH, and others) or in first quartile journals (Sensors, ESWA, Neurocomputing, and others) need to be expanded. 

R= Thank you very much, the suggestions and references have been updated.

Modern methods can be found on paperwithcode, on the SOTA page of a particular case. For example, if we consider the modern, well-known and publicly available corpus (https://paperswithcode.com/sota/sign-language-recognition-on-autsl), the best methods at the moment are STF+LSTM (for mobile devices - overlapping with the review), SAM-SLR, Ensemble - NTIS, MViT-SLR, FE+LSTM (for mobile devices - overlapping with the review). Therefore, it is appropriate for the authors to analyze the best and known methods. All this will show that the authors are aware of the best results achieved on large corpora. Otherwise, it turns out that the authors only described neural network models for gesture recognition and did not mention the best results of 2022-24. The methods build on MediaPipe and other approaches and describe their application on mobile devices (STF+LSTM and others), so these methods are popular in the research community and can perfectly complement the tables in the paper. As for other corpora, for example, the HaGRID case, recently presented at the WACV 24 conference, is new and there are no SOTA results for it yet. As for the review of Kinect-related work (not only the first version, but also the second version - which is already considered obsolete), a certain number of corpora have also been collected for this purpose in the field of gesture recognition, and studies have been carried out based on them (see data from TheRusLan corpus - highly rated LREC conference) and methods.

R= Thank you very much, the suggestions were attended, See references

  1. In this article, the authors should clearly categorize existing approaches and methods for automatic hand gesture recognition and sign language elements into four distinct categories: (1) gesture recognition using sensor-equipped gloves, (2) marker-based motion capture systems, (3) video-based motion capture hardware, and (4) computer vision and machine learning methods. This categorization is essential to provide a comprehensive overview of the field.

Thank you very much, the suggestions were attended, see lines: 213-221 and Tables 3, 4 and 5.

Reviewer 2 Report

Comments and Suggestions for Authors

The article presents a comprehensive overview of static and dynamic hand gestures, encompassing numerous related studies and viewpoints.Nevertheless, there are several areas that could be improved to enhance the overall quality of the piece.

1. While the article analyzes static and dynamic hand gestures, it should be more focus on the latest developments in this field, reflecting the current state of research.

2. In section 5.2 (Gestures detection techniques), the author should analyze both the advantages and disadvantages of the static and dynamic hand gesture detection techniques which are pointed out.

3. The current use of references is limited and predominantly relies on older sources. It is crucial to include recent articles that have contributed significantly to the field. Additionally, the number of references should be expanded to provide a more robust support for the arguments and discussions presented.

4. The resolution of the images used in the article is insufficient, affecting their clarity and readability.

5. There is a noticeable overlap and repetition of content between the first and third paragraphs.

Comments on the Quality of English Language

On page 13, line 289, there is a grammatical error in the sentence, "for In the end get a negative type contour." 

Author Response

Dear Reviewer
Thank you very much for your comments. We have tried to comply with each and every one of them.

  1. While the article analyzes static and dynamic hand gestures, it should be more focus on the latest developments in this field, reflecting the current state of research.

R=Thank you very much, the suggestions were attended, see lines: 213-221 and Tables 3, 4 and 5.

  1. In section 5.2 (Gestures detection techniques), the author should analyze both the advantages and disadvantages of the static and dynamic hand gesture detection techniques which are pointed out.

R=Thank you very much, the suggestions were attended, see lines 188-213.

  1. The current use of references is limited and predominantly relies on older sources. It is crucial to include recent articles that have contributed significantly to the field. Additionally, the number of references should be expanded to provide a more robust support for the arguments and discussions presented.

R= Thank you very much, the suggestions were attended, see lines: 213-221 and Tables 3, 4 and 5.

  1. The resolution of the images used in the article is insufficient, affecting their clarity and readability.

R= Thank you very much, the suggestions were attended. All the figures were improved

  1. There is a noticeable overlap and repetition of content between the first and third paragraphs.

R= Thank you very much, the suggestions were attended to. See lines 190-213.

Round 2

Reviewer 1 Report

Comments and Suggestions for Authors

The authors have improved the article. In this form the article can be accepted for publication.

Comments on the Quality of English Language

Minor editing of English language required.

Author Response

Dear Reviewer

Thank you very much for your comments, it has been of great support to improve the proposed work.

1. Minor editing of English language required.

R= Thank you very much, for the suggestions. MDPI's language editing service was used Id 80242. In addition, a Native English Speaker did a final review.                                          

Reviewer 2 Report

Comments and Suggestions for Authors

Thank you for taking the time to carefully address the comments and suggestions provided during the first round of review. I appreciate your diligent efforts in revising your manuscript.

Upon reviewing the revised version, I have some further suggestions that I believe would further enhance the quality and clarity of your paper. Please find them below:

1. In introduction, the author introduce the content of this article in the last paragraph. However there are some mistakes.The introduction incorrectly states that Section 3 covers haptic devices in VR. The content actually appears in Section 4.

2. In section 7.2(Conclusions), the author only mentions hand gesture application in VR, failing to analyze the combination of hand gestures and haptic devices.

Author Response

Dear Reviewer

Thank you very much for your comments, it has been of great support to improve the proposed work.

Upon reviewing the revised version, I have some further suggestions that I believe would further enhance the quality and clarity of your paper. Please find them below:

  1. In introduction, the author introduce the content of this article in the last paragraph. However there are some mistakes.The introduction incorrectly states that Section 3 covers haptic devices in VR. The content actually appears in Section 4.

R=Thank you very much, the suggestions were attended, see lines 53-58.

2. In section 7.2(Conclusions), the author only mentions hand gesture application in VR, failing to analyze the combination of hand gestures and haptic devices.

R=Thank you very much, the suggestions were attended, see lines 451-454.
